# Abg-CoQA: Clarifying Ambiguity in Conversational Question Answering

**Meiqi Guo**[1]                                                        MEIQI.GUO@PITT.EDU

**Mingda Zhang**[1]                                                        MZHANG@CS.PITT.EDU

**Siva Reddy**[2]                                                      SIVA.REDDY@MILA.QUEBEC

**Malihe Alikhani**[1]                                                          MALIHE@PITT.EDU

[1]*Department of Computer Science, University of Pittsburgh, Pittsburgh, PA 15260, USA*

[2]*MILA – Quebec Artificial Intelligence Institute, Montréal, QC H2S 3H1, Canada*

## Abstract

Effective communication requires the ability to identify ambiguities and request clarification of utterances. For machines to engage in a conversation, they need to learn to generate different forms of clarification questions. This paper aims at studying the extent to which the state of the art neural generation models can generate effective clarification questions in conversational question answering. We introduce Abg-CoQA, a novel crowdsourced dataset for clarifying ambiguities in conversational question answering systems. Our dataset contains 8,615 questions with answers where 994 questions are ambiguous. The conversational questions are about 3,968 text passages from five diverse domains which are pre-selected from the CoQA dataset. For ambiguous turns, we have collected the clarification questions and their answers. We evaluate strong language generation models and conversational question answering models on Abg-CoQA. The best-performing system achieves a F1-score of 23.6% on ambiguity detection; an accuracy of 56.0% on generating clarification question in human evaluation; and a F1 score of 40.1% on question answering after clarification, which is 35.1 points behind human performance (75.2%), indicating there is ample room for improvement.

## 1. Introduction

People naturally resolve ambiguities in conversation by asking context-dependent clarification questions [Clark and Brennan, 1991]. Although there has been a surge in research on conversational question answering [Choi et al., 2018, Reddy et al., 2019], few studies have explored ambiguity resolution and clarification. Question answering (QA) systems' goal is to return the answer to a question by searching for a response in a collection of documents [Rajpurkar et al., 2016, 2018, Seo et al., 2017, Wang et al., 2017]. Often these systems may need additional information in order to answer an user's query. Similar to human-human conversation, QA systems needs to identify and clarify ambiguities in order to be able to return the correct response.

In this paper, we introduce Abg-CoQA, a dataset for clarifying the **am**bi**g**uity in **Co**nversational **Q**uestion **A**nswering. The specific problem we address in this paper is on detecting ambiguities and generating the right kinds of clarification questions for resolving them. As Figure 1 shows, the model needs to first detect whether a question $(Q_i)$ in a conversation is ambiguous or not. Then, for ambiguous questions it needs to generate

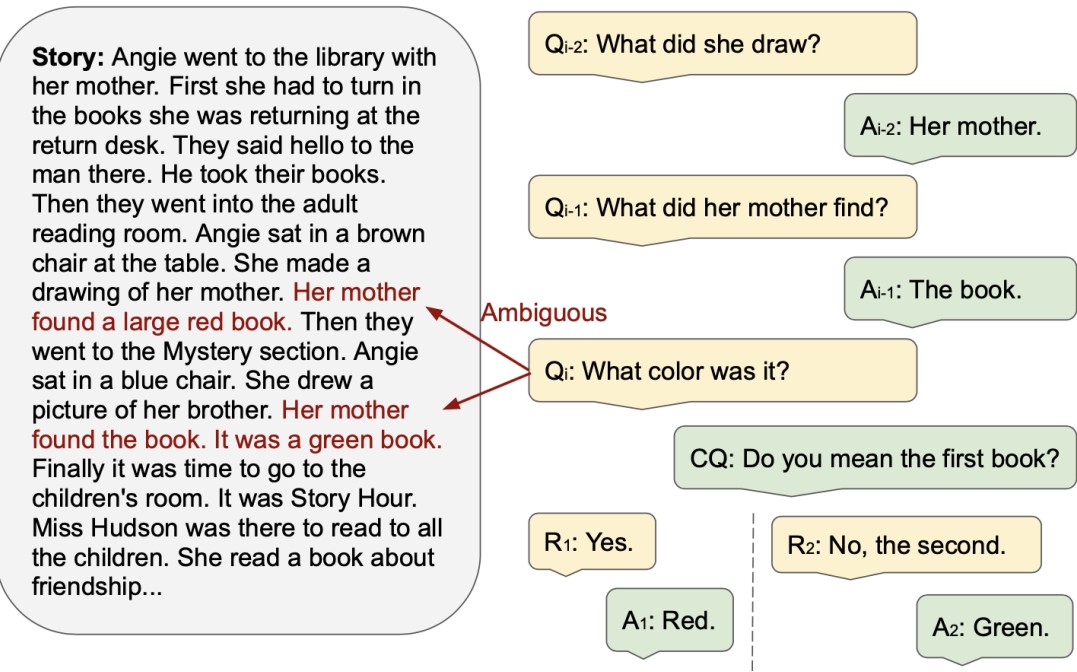

Figure 1: A conversation from the Abg-CoQA dataset.

a clarification question ($CQ$). Since there are in general several possibilities for answering $CQ$, it then needs to provide an answer ($A_k$) based on each possible clarification reply ($R_k$).

Our first objective is to study how and when ambiguities arise in conversational question answering. Our dataset covers four types of ambiguities (Table 2) in human conversations which are related to the two ambiguity types introduced in Ginzburg [1996]: 1) when the question's focus is ambiguous; 2) when there exist several possibilities to answer the question [Larsson, 2002]. The second objective is to enable the design of robust question answering systems that can handle ambiguities in longer interactions.

To summarize, Abg-CoQA has the following key characteristics: 1) it consists of 3,968 passages from five different domains and 8,615 conversational questions where 994 of them are ambiguous; 2) it covers four different ambiguity types and in most cases, the ambiguity is apparent after referring to the conversation history and researching all possible answers in the story; 3) each ambiguous question is clarified by a clarification question and several possible replies which lead to different answers to the originally ambiguous question.

Building on state-of-the-art conversational question answering and neural language generation models, we present several benchmarks in ambiguity detection, generation of clarification questions and answering questions which are disambiguated by the clarification. We demonstrate the limitations of these end-to-end models and discuss possible future work. Our code and data is publicly available at : https://github.com/MeiqiGuo/AKBC2021-Abg-CoQA.

## 2. Related Work

The ability to request clarification of utterance in dialogue has been extensively studied in dialogue research [Purver et al., 2003, Schlangen, 2004, Stoyanchev et al., 2014]. To the best of our knowledge, however, most of the existing conversational question answering benchmarks have either underestimated or ignored the possibility of the presence of ambiguities in conversational question answering.

Khalid et al. [2020], Rodríguez and Schlangen [2004] have combined cognitive modeling and discourse theories with reinforcement learning to choose the best clarification requests from a set of predefined strategies. Our focus, however, will be on exploring the potential of large neural language generation models in automatically generating the optimal clarification questions in conversational question answering.

Min et al. [2020] investigated ambiguity resolution in open-domain question answering through question rewriting. Our work, however, focuses on clarifying ambiguities in conversational question answering by interactively asking clarification questions [Traum, 1994, Kato et al., 2013].

Prior work have studied the types, subjects and effectiveness of clarification questions that users ask on the Stack Exchange community question answering platform [Braslavski et al., 2017, Rao and Daumé III, 2018]. Our work differs from these two studies in that conversational question answering is in multi-turn and the clarification has a direct impact on the answer. Aliannejadi et al. [2019], Zamani et al. [2020] study a sequence of clarification questions to refine intents of simple searching query. Saeidi et al. [2018] worked on natural language rules in conversational machine reading and showed the success of crowdsourcing in answering the ambiguous questions that require access to world knowledge. Xu et al. [2019] define similar tasks on clarifying ambiguity for knowledge-based question answering where the ambiguity types are only limited to entity reference and pronoun reference. We focus on resolving ambiguities that naturally arise in conversational question answering about a passage (Table 2).

## 3. Task Definition

Figure 1 depicts the Abg-CoQA task. Given a passage $P$ and a conversation $\{Q_{i-n}, A_{i-n},..., Q_{i-1}, A_{i-1}\}$ (where $n$ is the number of the conversation turns), the task is to clarify the ambiguity in the next question $Q_i$ if it is ambiguous. We consider three tasks.

**Ambiguity Detection.** Given a passage $P$ and a conversation $Q_{i-n}, A_{i-n},..., Q_{i-1}, A_{i-1}$, detect whether the next question $Q_i$ is ambiguous.

**Clarification Question Generation.** Given a passage $P$, a conversation $\{Q_{i-n}, A_{i-n},..., Q_{i-1}, A_{i-1}, Q_i\}$ where $Q_i$ is ambiguous, generate a clarification question $CQ$ which is helpful for disambiguating $Q_i$.

**Clarification-based Question Answering.** Given a passage $P$, a conversation $\{Q_{i-n}, A_{i-n},..., Q_{i-1}, A_{i-1}, Q_i, CQ, R_k\}$ where $Q_i$ is an ambiguous question, $CQ$ is a clarification question for $Q_i$, and $R_k$ is one possible reply to $CQ$, answer the question $Q_i$ which is no longer ambiguous based on the clarification. There should be different answers $A_{R_k}$ for every $R_k$.

## 4. Data Collection

We construct Abg-CoQA based on the CoQA dataset [Reddy et al., 2019]. Since most questions in the CoQA dataset are not ambiguous, we increase the ambiguity rate in our annotated corpus by 1) considering a partial conversation (keeping several previous conversational turns) rather than the full conversation; 2) pre-select probably ambiguous questions by using question answering models which are trained on CoQA dataset. We use Amazon Mechanical Turk (AMT) for crowdsourcing.

### 4.1 Collection Process

Given a story and a conversation (which is generally partial), annotators are asked to identify whether a question is ambiguous or not. If it is ambiguous, then provide a clarification question and all possible replies to it (could be one or several replies, see Figure 1). We also ask annotators to write all possible answers to the initial ambiguous question according to each clarification reply[1].

In order to ensure the annotation quality on the crowd-sourcing platform, we filter AMT workers by location (US, CA, IN only for making sure that workers are English speakers), assignment approval rate (>97%) and customized qualification tests (for making sure that workers understand the guidelines).

### 4.2 Ambiguous Question Pre-Selection

We pre-select samples which are likely to be ambiguous from CoQA. According to Reddy et al., all models succeed at leveraging history but the gains quickly drop beyond one previous turn. They have the same observation with human performance: given two history turns, human performance reaches up to almost same as given the full history. This suggests that most questions in a conversation have a limited dependency within a bound of two turns. Therefore, in our task, we only provide one or two history turns, which decreases the annotators' work load (shorter conversation) and potentially increases the ambiguity rate of questions (some questions may have longer dependency than 2 turns).

We pre-select questions which get a wrong answer given an incomplete conversation history but are answered correctly given the full history. The intuition is that those questions turn to be ambiguous because of the shorter conversation history rather than the inherent difficulty for answering the question itself. We first train a baseline model which has a BERT-based architecture with answer verification on the CoQA training dataset, given the full conversational history as input [Wu et al., 2019]. Then we select a sample if the model prediction given an incomplete history as input is greatly worse than given the full history. With this pre-selection process, we construct our corpus to be annotated.[2]

---

1. Refer to Appendix A for examples of the annotation interface.
2. Besides CoQA, we also considered QuAC [Choi et al., 2018] as our data source since it is also a conversational question answering dataset based on context. We follow the same process for pre-selecting potentially ambiguous questions using a BERT-based model with history attention mechanism [Qu et al., 2019]. However, our pilot study on 50 samples shows that the ambiguity rate is very low – 2%. We don't include QuAC in our work because of its low annotation efficiency.

| Domain | Total | | Ambiguous | | Abg rate |
|---|---|---|---|---|---|
| | #P | #Q | #P | #Q | %Q |
| Children's Stories | 296 | 636 | 71 | 90 | 14.2 |
| Literature | 991 | 2201 | 180 | 203 | 9.2 |
| Mid/High School Exams | 955 | 2172 | 186 | 226 | 10.4 |
| News | 909 | 1894 | 206 | 246 | 13.0 |
| Wikipedia | 817 | 1712 | 198 | 229 | 13.4 |
| TOTAL | 3968 | 8615 | 841 | 994 | 11.5 |

Table 1: Distribution of data numbers and ambiguity rates with respect to the domains in Abg-CoQA. In the column of *Total*, #P and #Q respectively mean the total number of passages and conversational questions; in the column of *Ambiguous*, #P means the number of passages where at least one ambiguous question is identified in the conversation and #Q means the number of ambiguous questions; in the column of *Abg rate*, the rate of ambiguous questions with respect to all questions is computed.

With respect to the data splitting, we follow the same way as CoQA. For each source domain (*e.g.*, Children's Stories), we split the data such that there are 100 passages in the development set, 100 passages in the test set, and the rest in the training set.

## 5. Data Analysis

The final dataset contains 3,968 passages and 8,615 questions, where 994 questions are annotated as ambiguous.

**Domain Distribution.** Table 1 shows the distribution of passages and questions with respect to the source domains of CoQA. We observe that the domain of Literature has the lowest ambiguity rate and the domain of Children's Stories has the highest. This meets our intuition that language uses in Literature are more precise therefore there is less ambiguity in the conversation; in contradictory, Children's Stories is generally informal.

**Types of Ambiguity.** Table 2 shows a breakdown of the types of ambiguity in Abg-CoQA. We define a taxonomy with four categories, including ambiguity in coreference resolution, event references, time-dependency, and answer types. According to the two ambiguity types introduced in Ginzburg [1996], the ambiguity in coreference resolution is about the question focus and the ambiguity in answer types is about the answering possibilities; the ambiguity in event references and time-dependency cover the both ambiguity types. In comparison to Min et al. [2020], who studies ambiguity in open-domain questions, our corpus contains one new ambiguity type – coreference resolution which is an inherent challenge in conversations. In addition, different from open-domain questions where more than a tier have the ambiguity in event references, it is actually a minor class in conversational questions since requested events are under the scope of the given story. In most cases, ambiguity is not apparent from the prompt question alone, but only after referring to the conversation history and researching all possible answers in the story.

**Clarification Strategies.** We classify the clarification questions in three types: More Information, Selection and Check. Our taxonomy follows Kato et al. [2013]'s work which

| Type | Example |
|---|---|
| Coreference resolution (49%) | **Story**: ... Out of Africa(1985). Meryl is Karen Blixen, a Danish woman living in Kenya. The story follows Karen's attempts to run a coffee plantation and her love affair with ... 
 **$Q_{i-2}$**: What was her next movie? 
 **$A_{i-2}$**: Out of Africa. 
 **$Q_{i-1}$**: What type of character did she play? 
 **$A_{i-1}$**: Danish woman. 
 **$Q_i$**: What did she do? 
 **Clarification question**: By "she" are you referring to Meryl Streep or her character? 
 **Clarification reply #1**: I am referring to her character in Out of Africa. 
 **$A_i$ #1**: Her character, Karen, attempts to run a coffee plantation and has an affair. 
 **Clarification reply #2**: I mean Meryl Streep. 
 **$A_i$ #2**: She is an actress who has worked in theatre, film, and television. |
| Time-dependency (23%) | **Story**: ... One of us grabbed a big wheel and rode it down the steep driveway into the street. Greg and I did it several times until the last time. The car hit him on the head. My brother and I both ran screaming just yelling for help and crying... 
 **$Q_{i-2}$**: Did anybody actually see the accident happen? 
 **$A_{i-2}$**: Yes. 
 **$Q_{i-1}$**: Who saw it? 
 **$A_{i-1}$**: My brother and I. 
 **$Q_i$**: What was everyone doing? 
 **Clarification question**: Do you mean before the accident? 
 **Clarification reply #1**: Yes. 
 **$A_i$ #1**: Riding a big wheel down the driveway into the street. 
 **Clarification reply #2**: No, after the accident. 
 **$A_i$ #2**: We ran screaming and yelling for help and crying. |
| Answer types (16%) | **Story**: ... His ninth-grade English class for boys centers on books. "The novels they're reading now, are very manly novels. They're novels that deal with the arrogance of man and the pride of man." One of those books, for example, is "The Call of the Wild"... 
 **$Q_{i-2}$**: Who does he teach? 
 **$A_{i-2}$**: Boys. 
 **$Q_{i-1}$**: What are his pupils doing? 
 **$A_{i-1}$**: They're reading. 
 **$Q_i$**: What? 
 **Clarification question**: The type of book or an example of a book they're reading? 
 **Clarification reply #1**: The type of book. 
 **$A_i$ #1**: Novels that deal with the arrogance of man and the pride of man. 
 **Clarification reply #2**: An example of a book. 
 **$A_i$ #2**: "The Call of the Wild". |
| Event references (12%) | **Story**: ... Dallas police named the suspected shooter, though CNN is not identifying him yet since he's a minor. The teen turns 18 in May, police said... 
 **$Q_{i-1}$**: How old? 
 **$A_{i-1}$**: 17. 
 **$Q_i$**: Was he identified by name? 
 **Clarification question**: Do you mean identified by whom? 
 **Clarification reply #1**: I mean by Dallas police. 
 **$A_i$ #1**: Yes. 
 **Clarification reply #2**: I mean by CNN. 
 **$A_i$ #2**: No. |

Table 2: Breakdown of the types of ambiguity in 50 random samples from ambiguous cases.

classifies clarification requests of users in six categories, however, we only consider three types among them since we focus on the clarification strategy rather than the user intent. *Check* aims to confirm a hypothesis corresponding to the ambiguity (*e.g.*, the second example in Table 2); *Selection* aims to request an answer from two or more possibilities about the ambiguity (*e.g.*, the first and third example in Table 2); *More Information* directly asks for further details for clarifying the ambiguity (*e.g.*, the last example in Table 2). We compute the distribution of clarification strategies in 50 random ambiguous items: *Check*, *Selection* and *More Info* respectively occupies 47%, 43% and 10% of samples. It shows that people prefer verifying their hypotheses (*e.g.*, *Check*, *Selection*) rather than asking open questions (*e.g.*, *More Info*) for clarifying the ambiguity.

In addition, we compute the distribution of the number of clarification replies in all ambiguous items. Most clarification questions have two different replies (74%); 11% of them have only one reply[3]; and 15% of them have more than two replies.

## 6. Models

To set initial performance levels on Abg-CoQA, we present a baseline model for each task. These tasks cover both the conversational question answering and language generation.

**Ambiguity Detection.** We formulate this task as the traditional question answering task by adding "ambiguous" as a possible prediction output. We consider two extraction-based models which have shown promising results for generating conversational responses on the CoQA dataset as the baseline models for this task. Our baseline models are respectively build upon BERT [Devlin et al., 2019] and XLNET [Yang et al., 2019] plus prediction heads for each answer type[4] (respectively called BERT+AnsType [Wu et al., 2019] and XLNET+AnsType[5]).

In order to take the *ambiguity* of questions into consideration, we append the "ambiguous" token at the end of the input passage so that models are able to extract it as the prediction result[6]. Therefore, the input to the model is the passage appended by "ambiguous", the conversation history and the question, and the expected output is the specific "ambiguous" token when the question is ambiguous; or the original response to the question when it is not ambiguous.

**Clarification Question Generation.** We fine tune a strong model for text generation – BART [Lewis et al., 2020] on our corpus as the baseline model for generating clarification questions. We append the given conversation history and the current question to the text passage and feed it into BART. The expected output is the clarification question. Since the ambiguous samples are in a small amount (*i.e.*, 1k), we also consider adding an additional fine-tuning prior to this clarification question generation task: generating the next question

---

3. Those cases mainly come from one clarification reply is highly possible than others. When annotators find the question is ambiguous and write those clarification questions, they aim to confirm their hypothesis (very likely to be true) rather than seeking an answer.

4. Answer type could be yes/no/unknown/extraction. For most cases, the prediction answers can be extracted from the input passage by predicting the start and end positions. The specific token may not exist in the passage if the answer is yes/no/unknown.

5. We adapt the XLNET extension toolkit: https://github.com/stevezheng23/xlnet_extension_tf

6. With the answer type of extraction.

given the conversation history on the CoQA dataset (excluding the test set of Abg-CoQA) in order to make the model learn to generate common conversational questions.

**Clarification-based Question Answering.** We formulate this task as the original conversational question answering task by considering the clarification as the previous conversation history. In this task, we append the clarification question and one possible reply to the passage and the conversation as the input sequence to the model. The expected output is the answer to the originally ambiguous question based on the clarification. We consider three different types of models as our baseline: BERT+AnsType [Wu et al., 2019] and XL-NET+AnsType which are also used for the task of ambiguity detection and a generative model GPT-2 [Radford et al.] for zero-shot prediction. For BERT+AnsType and XL-NET+AnsType, we first pre-train them on CoQA then fine-tune on Abg-CoQA by adding the clarification into the conversation.

## 7. Evaluation

We first evaluate the inter-rater agreement of annotations and discuss the subjectivity of clarifying ambiguity. Then we report the performance of several state-of-the-art models on each task as well as the take-away observations.

### 7.1 Evaluation Metric

For answer generation, we use the same metric as CoQA: macro-average F1 score of word overlap. For computing a model's performance, each individual prediction is compared against $n$ human answers resulting in $n$ F1 scores, the maximum of which is chosen as the prediction's F1. For each question, we average out F1 across these $n$ sets, both for humans and models. We follow the same way as CoQA for fixing the bias when computing human performance. In our evaluation on clarification-based question answering, $n$ is equal to 3.

For detecting ambiguity, we compute the precision, recall and F1 as the evaluation metric on the two-class classification. We also report the macro-average F1 on answer generation ($n = 1$) since predicting "ambiguous" is formulated as extracting an answer span.

For clarification question generation, we use BLEU scores as the main metric with the human-annotated clarification question as the gold standard. In addition to measuring the similarity with BLEU, we manually evaluate whether the generated question is reasonable and helpful for clarifying the existing ambiguity.

### 7.2 Inter-rater Agreement

For measuring the inter-rater agreement on whether a question is ambiguous or not, we compute the Cohen's Kappa score on 68 randomly selected samples which are annotated by two Amazon Turk workers. The Cohen's Kappa score on the ambiguity detection is equal to 0.26 (a fair agreement), indicating the task is subjective and challenging.

For measuring the inter-rater agreement on clarification questions, we ask a second Amazon Turk worker for writing a clarification question (and all possible replies) on ambiguous samples of the development and test sets. We compute the macro-average F1 score of word overlap and the BLEU scores. The F1 score is 45.3% and the BLEU-4 score is 21.9 on the test set (more BLEU scores are reported in Table 4).

| Model | Question Answering (F1) | | | | | | Ambiguity Detection | | |
|---|---|---|---|---|---|---|---|---|---|
| | Child. | Liter. | School | News | Wiki | TOTAL | Prec. | Recall | F1 |
| BERT+AnsType | 30.3% | 36.0% | 31.4% | 30.3% | 28.9% | 31.4% | 19.0% | 26.6% | 22.1% |
| XLNET+AnsType | 41.8% | 43.3% | 52.7% | 40.8% | 48.3% | 45.5% | 30.0% | 19.5% | 23.6% |

Table 3: Results on ambiguity detection.

| Model | BLEU-1 | BLEU-2 | BLEU-3 | BLEU-4 |
|---|---|---|---|---|
| Human Performance | 40.8 | 31.6 | 26.2 | 21.9 |
| fine-tune BART on Abg-CoQA | 36.5 | 23.6 | 16.8 | 11.9 |
| fine-tune BART on CoQA + Abg-CoQA | 38.2 | 24.9 | 18.1 | 13.3 |

Table 4: Results on generating clarification questions.

For the clarification-based question answering, we ask three annotators to answer each ambiguous question in the test set after considering the clarification. The macro-average F1 score of these three annotations is equal to 75.2%.

### 7.3 Subjectivity of Clarifying Ambiguity

The inter-rater agreement score on ambiguity detection indicates that it is a subjective task. In order to verify if this subjectivity does not hurt the quality of annotations, we evaluate them through the agreement on answering ambiguous questions. We randomly select 100 ambiguous cases (based on the original annotation) in development set and ask three annotators to write an answer to each ambiguous question. We compute the macro-average F1 score of word overlap as a way to measuring the human agreement on answering ambiguous question, which is 65.3%. Then we compare it with the F1 score of the clarification-based question answering which is 75.2%. It shows that the clarification increases 10 points for the inter-rater agreement (measured by F1 score) on answering ambiguous questions, which demonstrates that those questions are indeed ambiguous and the clarification is effective.

How to formulate a clarification question is also subjective since there are different clarification strategies (Section 5) and annotators may use different expressions for clarifying the same ambiguous doubt. For verifying this hypothesis, we randomly select 50 samples and manually compare the difference between the two clarification questions for each sample. We observe that more than a half (57%) of them ask additional information for the *same* ambiguous doubt by using *different* expressions or clarifying strategies, which supports our hypothesis. Besides, only 17% of them are exactly the same and 26% of them target different ambiguity.

### 7.4 Results and Discussion

We report the experimental results of baseline models on our defined three tasks. The detailed experimental setting is introduced in Appendix B.

#### 7.4.1 AMBIGUITY DETECTION.

*It is a challenging task for the state-of-the-art question answering models.*

| Model | Coreference resolution | Time-dependency | Answer types | Event references | **TOTAL** |
|---|---|---|---|---|---|
| fine-tune BART on Abg-CoQA | 31.8% | 0.0% | 30.0% | 45.5% | 30.0% |
| fine-tune BART on CoQA + Abg-CoQA | 54.5% | 71.4% | 40.0% | 63.6% | 56.0% |

Table 5: Human evaluation on the accuracy of generated clarification questions.

| Model | Children's Sto. | Literature | Mid/High Sch. | News | Wikipedia | **TOTAL** |
|---|---|---|---|---|---|---|
| Human | 74.6% | 73.0% | 76.2% | 76.5% | 74.7% | 75.2% |
| XLNET+AnsType | 29.6% | 32.2% | 44.4% | 47.8% | 42.2% | 40.1% |
| BERT+AnsType | 30.2% | 31.4% | 36.6% | 49.1% | 40.9% | 38.7% |
| GPT-2 zero-shot | 14.5% | 7.0% | 11.5% | 17.4% | 6.9% | 11.8% |

Table 6: F1 scores on answer prediction after the clarification.

Since we consider ambiguity detection in the question answering setting, we report both the performance on answer prediction and on ambiguity detection in Table 3. With respect to the performance on question answering, the model trained on Abg-CoQA acheives an F1 score equal to 31.4% for BERT-based and 45.5% for XLNET-based. It is not surprising because the baseline model failed on these samples when trained on CoQA (samples were selected by the pre-selection process; original average F1 score on them is only 3.8%). The best ambiguity detection performance gets a F1 score of 23.6%, which reveals the challenging of this task. We observe that the XLNET+AnsType baseline has a higher precision while the BERT+AnsType model has a higher recall.

### 7.4.2 CLARIFICATION QUESTION GENERATION.

*Pre-training on CoQA on the question generation task helps but doesn't solve it. Human evaluation is necessary besides automatic metrics.*

Results for the clarification question generation task is shown in Table 4. For determining whether the generated question is reasonable/helpful for clarifying the existing ambiguity, we randomly select 50 samples in the test set and manually evaluate them. We report the accuracy in Table 5 for both the general performance and each ambiguity type. According to the BLEU metrics, pre-training on CoQA on the question generation task slightly improves the performance (by around 2 points) comparing to directly fine-tuning BART on Abg-CoQA. However, human evaluation shows that the improvement is greatly significant: gaining 26% points of accuracy. This observation reveals that though BLEU metrics are correlated with human evaluation, the scaling is not a straightforward signal for the comparison in this case. The best performing model gets an accuracy score of 56%, which shows that there is still a gap comparing to the human performance. By looking into each ambiguity type, we observe that models are generally good at clarifying the ambiguity of event references, while the ambiguity of answer types is the most difficult for models. Interestingly, pre-training BART on CoQA significantly helps the model generating clarification questions when the ambiguity type is time-dependency (from 0.0% accuracy to 71.4%).

### 7.4.3 CLARIFICATION-BASED QUESTION ANSWERING.

*It demands more reasoning for models than common question answering. The model built upon XLNET achieves the best performance but still largely behind human.*
Results for the answer prediction after the clarification is shown in Table 6. The model built upon XLNET achieves the best performance, however, still 35.1% behind the human performance, which shows that our task brings a new challenge to the question answering community. Even though samples in Abg-CoQA were pre-selected by the BERT-based baseline model, we don't see a great difference of performance between BERT and XLNET on this task. It demonstrates that the task is not biased towards the pre-processing of the BERT-based model. We also run GPT-2 as a representative of generative models on zero-shot prediction. Its performance decreases more than 40 points comparing to the reported F1 score (55%) on CoQA [Radford et al.].

We conduct an error analysis and find that existing strong models on standard conversational question answering tasks actually can't correctly answer the question based on different clarification replies. For example in Fig 1, models always predict "green" no matter the clarification reply to "Do you mean the first book?" is "Yes" or "No, the second". It reveals that current models may be saturated to the training distribution rather than truly understand the conversation.

## 8. Conclusions

Our empirical study and computational experiments show that identifying and resolving ambiguities in a conversation is rather a challenging task even for people. We discuss the success and limitations of the state-of-the-art end-to-end neural models and present a dataset that can be used for further investigating solutions for managing ambiguities in conversational question answering. A direct application of pre-trained large models, *e.g.* BERT, BART, is not able to identify ambiguities or to generate clarification questions for resolving the ambiguity. Future research developing on Abg-CoQA models may include modeling uncertainty as a signal of ambiguity and generating clarification questions in several steps, such as first identifying the ambiguity type, then generating a contextual phrase which targets the ambiguity, finally formulating the complete question.

## 9. Ethics

We have not worked with sensitive or personal data. The presented dataset is fully anonymized. Our work relies on pretrained models. These models are known to reproduce and magnify societal bias that is presented to them in the training process. Moreover, like other deep learning methods, the presented models are likely to perform better for content that is better represented in training, leading to further bias against marginalized groups. We hope that general methods to mitigate harms from ML bias can address these issues.

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

## Appendix A. Examples of Annotation Interface

Figure 2 and Figure 3 show examples of the interface for AMT annotators.

## Appendix B. Experimental Setup

For the BERT-based model, we adapt the SogouMRCToolkit[7] to our dataset and use its setting: batch size of 6, number epoch of 10, warm-up proportion of 0.1. We use the uncased-base BERT model as the backbone. The models are optimized using Adamax,

---

7. https://github.com/sogou/SogouMRCToolkit

**Before you accept this HIT, please read the following instructions carefully.**

**Instructions** (click to collapse)

- Here we are interested in **clarifying ambiguous questions in a conversation which discusses a story**. You need to decide whether a question is ambiguous or not. If it is ambiguous, then you need to conduct a conversation for clarifying it.
- **Please read this tutorial carefully before continuing on the HIT.**
- Please note that this is **NOT a survey** about your personal opinions. You should try to **answer these questions from a general viewer's perspective**.
- The automatic approval time for this HIT is 24 hours.
- If you have any comments or encounter any problems, you can email us at ambiguousdialog@gmail.com

**If you have read our instructions and tutorial and would like to continue, please answer the following questions.**

**Story:**
The term Hispanic (or) broadly refers to the people, nations, and cultures that have a historical link to Spain. It commonly applies to countries once owned by the Spanish Empire in the Americas (see Spanish colonization of the Americas) and Asia, particularly the countries of Hispanic America and the Philippines. It could be argued that the term should apply to all Spanish-speaking cultures or countries, as the historical roots of the word specifically pertain to the Iberian region. It is difficult to label a nation or culture with one term, such as "Hispanic", as the ethnicities, customs, traditions, and art forms (music, literature, dress, culture, cuisine, and others) vary greatly by country and region. The Spanish language and Spanish culture are the main distinctions.

"Hispanic" originally referred to the people of ancient Roman Hispania, which roughly comprised the Iberian Peninsula, including the contemporary states of Spain, Portugal, Andorra, and the British Overseas Territory of Gibraltar.

The term "Hispanic" derives from Latin "Hispanicus" ('Spanish'), the adjectival derivation of Latin (and Greek) "Hispania" ('Spain') and "Hispanus"/"Hispanos" ('Spaniard'), ultimately probably of Celtiberian origin. In English the word is attested from the 16th century (and in the late 19th century in American English).

**Conversation:**
...
$Q_{i-2}$: What are the primary distinctions?
$A_{i-2}$: the Spanish language and culture

$Q_{i-1}$: Was Andorra part of Roman Hispania?
$A_{i-1}$: yes

**After reading the above text, you need to answer the following questions.**

The next question $Q_i$ after the above conversation is "Name another area that was part of that region.". Is it an ambiguous question?
- ○ Ambiguous (you are not sure what the question is asking.)
- ◉ Non-ambiguous (the question is clear.)

Please write the answer to the question $Q_i$ according to the story and the conversation history.

[                                                                    ]

[ submit ]

Figure 2: An example of the annotation interface when annotator selects *Non-ambiguous*. The interface updates with the chosen options.

with a learning rate of 3e-5. For the XLNET-based model, we adapt the XLNET extension toolkit[8]. The batch size is 8, the number of training steps is 6000. The model is optimized using Adam, with a learning rate of 3e-5. For the GPT-2, we follow its reported setting on the zero-shot CoQA task [Reddy et al., 2019]: add "Q" before each conversational question and clarification question and "A" before each answer as well as the end of the input sequence. For fine-tuning the BART model, we use the Fairseq toolkit [Ott et al., 2019]. We use the pre-trained large model, with Adam optimizer and learning rate of 3e-05. The total number of training steps is 2000 and the number of warm-up steps is 50. We follow the default decoding strategy by setting the beam size as 5 and removing duplicated trigrams in beam search. For generative models, *i.e.*, BART and GPT-2, we only consider

---

8. https://github.com/stevezheng23/xlnet_extension_tf

**Before you accept this HIT, please read the following instructions carefully.**

**Instructions** (click to collapse)

- Here we are interested in **clarifying ambiguous questions in a conversation which discusses a story**. You need to decide whether a question is ambiguous or not. If it is ambiguous, then you need to conduct a conversation for clarifying it.
- **Please read this tutorial carefully before continuing on the HIT.**
- Please note that this is **NOT a survey** about your personal opinions. You should try to **answer these questions from a general viewer's perspective**.
- The automatic approval time for this HIT is 24 hours.
- If you have any comments or encounter any problems, you can email us at ambiguousdialog@gmail.com

**If you have read our instructions and tutorial and would like to continue, please answer the following questions.**

**Story:**
The term Hispanic (or) broadly refers to the people, nations, and cultures that have a historical link to Spain. It commonly applies to countries once owned by the Spanish Empire in the Americas (see Spanish colonization of the Americas) and Asia, particularly the countries of Hispanic America and the Philippines. It could be argued that the term should apply to all Spanish-speaking cultures or countries, as the historical roots of the word specifically pertain to the Iberian region. It is difficult to label a nation or culture with one term, such as "Hispanic", as the ethnicities, customs, traditions, and art forms (music, literature, dress, culture, cuisine, and others) vary greatly by country and region. The Spanish language and Spanish culture are the main distinctions.

"Hispanic" originally referred to the people of ancient Roman Hispania, which roughly comprised the Iberian Peninsula, including the contemporary states of Spain, Portugal, Andorra, and the British Overseas Territory of Gibraltar.

The term "Hispanic" derives from Latin "Hispanicus" ('Spanish'), the adjectival derivation of Latin (and Greek) "Hispania" ('Spain') and "Hispanus"/"Hispanos" ('Spaniard'), ultimately probably of Celtiberian origin. In English the word is attested from the 16th century (and in the late 19th century in American English).

**Conversation:**
...
$Q_{i-2}$: What are the primary distinctions?
$A_{i-2}$: the Spanish language and culture

$Q_{i-1}$: Was Andorra part of Roman Hispania?
$A_{i-1}$: yes

**After reading the above text, you need to answer the following questions.**

The next question $Q_i$ after the above conversation is "Name another area that was part of that region.". Is it an ambiguous question?
- ● Ambiguous (you are not sure what the question is asking.)
- ○ Non-ambiguous (the question is clear.)

Please write a clarification question below. A clarification question is for learning what the ambiguous question is exactly asking.

[                                        ]

Please write an answer to your clarification question.

[                                        ]

According to the clarification answer you just wrote, please answer the ambiguous question $Q_i$ ("Name another area that was part of that region.").

[                                        ]

Is there another possible answer to your clarification question?
- ● Yes
- ○ No

Please write an answer to your clarification question.

[                                        ]

According to the clarification answer you just wrote, please answer the ambiguous question $Q_i$ ("Name another area that was part of that region.").

[                                        ]

Is there another possible answer to your clarification question?
- ○ Yes
- ○ No

[submit]

Figure 3: An example of the annotation interface when annotator selects *Ambiguous*. The interface updates with the chosen options.

the generated first sentence for evaluation since the number of tokens in generated text is in general defined larger than the ground truth.

