# OpenReview forum: "Abg-CoQA: Clarifying Ambiguity in Conversational Question Answering"
_AKBC.ws/2021/Conference — AKBC 2021_

### Official Review · Reviewer_fbRa · 2021-07-12
**A solid dataset paper**

**Rating:** 7
**Confidence:** 4

**Review:**

In general, I think this is a solid dataset paper that targets a very important research problem: Dealing with the unanswerable questions in dialogue systems. But still, I have some questions/concerns. Hope that the authors can answer them.

1. I think the logic in your third goal is not very clear. Can you elaborate more on how performing robustly on the same question according to different clarification turns can show that models are generating responses based on the "true understanding of the context" or superficial features? I think this argument is not well supported by the paper.

2. It seems weird to use different metrics for generating responses for generating clarification questions and answering the questions. Have you considered using a unified evaluation metric such as generation metrics? Because in a real application, it is hard to get the candidates, right?

3. It seems like the dataset is still too small to train a good model. Based on your experimental results, state-of-the-art models are learning almost nothing. What are your insights into this?

4. Do you have any support (e.g., linguistic theory) to show that the classification types are necessary and complete?

5. From the IAA, it seems like detecting ambiguous questions is still very subjective. Do you think this task is well defined? Or how can we improve?

---

> ### Author Response · Authors · 2021-07-31
> **Response to R1**
>
> [Third goal logic] When CoQA was built, annotators read over a passage and then come up with a sequence of question-answer pairs. Those conversations may follow some intuitive features, such as the sequence of answers following the order in the passage; the answer appears in the sentence which is highly overlapped with the question, etc. Therefore, even though a model makes good predictions, it is difficult to isolate whether the model learns the real reasoning or captures some superficial patterns without further investigation. Our dataset can handle this issue: for the same question, different clarification turns lead to different answers. If a model only learns from superficial features, then the model will fail on our task, e.g., always predict the same answer even with different clarification replies. In addition, the clarification turn demands more complex reasoning for models because it is used for clarifying ambiguity which is not as obvious/easily-answered as a standard information-seeking conversation. Our claim is also consistent with our findings in Section 7.3.
>
> [Evaluation metric] The clarification question generation is a real generation problem, and the question answering task is actually formulated as a span extraction task (refer to Section 6). We use F1 score for evaluating the question answering tasks since it is a commonly-used metric and can be easily compared with other tasks (e.g., CoQA). And BLUE is a standard metric for evaluating the generating performance.
>
> [Insights on poor performance] We think the low performance mainly comes from the challenge of these tasks. Our tasks require complicated reasoning for models. Providing more data might boost the performance, however, in order to solve the problem researchers need to build new models which can clarify ambiguity with a strong reasoning capacity.
>
> [Theory support] As far as we know, there is no prior work which investigates different ambiguity types in conversational question answering. We are the first to explore it and the proposed taxonomy is one of our contributions. This taxonomy helps to understand different kinds of existing ambiguity and can be potentially beneficial to model building. However, we are not sure whether our taxonomy is complete for other general scenarios since it is summarized from our collected data.
>
> [Subjectivity of ambiguity detection] We agree that the task is subjective. It is because of the intrinsic nature of the ambiguity problem itself. For subjective problems, one way is to have more annotations (e.g. 10 annotators) and compute an average as the ambiguity level (a similar previous work [1] on emotion annotation since the emotion is also subjective). We didn’t follow it in this work since an absolute binary label is preferred and our collection achieves a fair agreement with Cohen’s Kappa equal to 0.26 [2]. Further work can be done for measuring the ambiguity level.
>
> [1] Yang J, Sun M, Sun X. Learning visual sentiment distributions via augmented conditional probability neural network[C]//Thirty-first AAAI conference on artificial intelligence. 2017.
>
> [2] Landis J R, Koch G G. The measurement of observer agreement for categorical data[J]. biometrics, 1977: 159-174.

---

### Official Review · Reviewer_918r · 2021-07-13
**A new dataset designed to address ambiguity in conversation**

**Rating:** 4
**Confidence:** 4

**Review:**

This work presents a new dataset designed to address ambiguity in conversation. It is well-motivated and presented, and follows most of the standard procedures expected for a dataset paper. My main concern is the low ambiguity rates, with only 994 of the
8615 questions found to be ambiguous. Even given the considerable data efficiency improvements of pre-trained LMs, I suspect this might be too little data for the models to learn anything useful, especially about something as challenging as ambiguity detection.

I also wonder if the pre-selection process might be limiting the annotation to certain types of questions only. It would be interesting to see an analysis of the ambiguity rates and challenges posed by questions with and without pre-selection.

The analyses by domain distribution and ambiguity types are both very detailed and insightful.

I am also slightly concerned about the dataset quality. A Cohen's kappa inter-rater reliability of 0.26 for question ambiguity detection is quite low, and it is difficult to isolate whether this is due to poor annotator quality or the ambiguity of the task itself (which could very well be expected given the nature of the task) without additional investigation. The authors present the answering of ambiguous questions as an attempt to make this more objective, although I believe it is investigating something slightly different i.e. whether ambiguous questions can be answered, rather than how well questions can be classified as ambiguous. In terms of evaluation of ambiguity detection, I'd also be interested to see how the other models perform as a single baseline for this sub-task is a bit limited.

Further investigation into why adding the CoQA questions to the training set for the generator makes things slightly worse would also be interesting.

Overall, I like the general direction of this work, but consider the amount of data collected to be insufficient (particularly with regards to the number of questions deemed to be ambiguous). I fully acknowledge the challenges of collecting data at scale, but presenting a dataset designed to investigate ambiguity clarification with < 1k such examples is just too small for my reaction to any experimental results not to be "but what if we gave the models more data?". I also have slight concerns about the dataset quality, and also consider the baselines to be quite limited.

Questions for authors:
- Are the data splits consistent with CoQA? i.e. are questions pertaining to the same passages in the dev set of CoQA also in the dev set of Abg-CoQA?
- You mentioned a customised qualification test to improve annotator quality, could you expand on that? What did it involve? Was its completion a strict requirement for working on the task?
- With regards to Fig 2b, where do the 11% with only one clarification reply come from? I would expect that an ambiguous question would have at least 2 possible references in the passage. For these examples, is this essentially the case but an alternative clarification reply has not been annotated in the dataset?
- For the generative models, what decoding strategy did you use?
- More of a pointer than a question, but I see many similarities in way the task is structured to ShARC (https://arxiv.org/abs/1809.01494) that might be worth discussing

---

> ### Author Response · Authors · 2021-07-31
> **Response to R2**
>
> [Concern on low ambiguity rate] A low ambiguity rate is expected since most of the human conversation is not ambiguous. The unbalance between ambiguous/non-ambiguous in our dataset is not an issue, given most abnormal/atypical detection tasks face the same situation. We agree that the task is challenging, however, the challenge mainly comes from the intrinsic property of the ambiguity detection rather than the data size. In addition, a total size of 8.6k is reasonable for the models to learn useful clues.
>
> [Concern on the pre-selection process] The pre-selection process is necessary since it helps to increase the ambiguity rate. Without the pre-selection, we would need to annotate much more samples in order to have 1k of them ambiguous (resulting in more annotation cost). We agree that the investigation of the potential limitation of the pre-selection is interesting and it could be done as future work. According to our analysis on the four different ambiguity types in Section 5 and the experimental results in Section 7.3, we argue that the pre-selection process doesn’t have a strong limitation on our collected data.
>
> [Concern on IAA] First, a Cohen’s Kappa of 0.26 is fair agreement [1]. Besides measuring the inter-rater agreement, we report the human agreement of answering ambiguous questions because a question is ambiguous if it makes several interpretations plausible. The macro-average F1 score of word overlap between three annotations is 65.3%, which is 10 points behind 75.2% F1 for answering ambiguous questions after the clarification. It shows that our annotation makes sense and works as expected. On the other hand, we control the annotation quality by providing clear coding manual, filtering workers by qualification tests and designing a specific annotation pipeline for quality control (i.e., if a worker annotates a question as ambiguous, then he/she needs to provide a clarification question and replies; if annotated as non ambiguous, then he/she needs to answer it). All the effort makes our collected dataset reliable.
>
> [Baseline for ambiguity detection] We added a second baseline model based on XLNET and reported the experimental result in Table 3. It performs similarly to the BERT-based baseline.
>
> [Question on data split] No, the data split is not consistent with CoQA. Because we want to make sure that the dev and test set contain 100 passages each.
>
> [Question on qualification test] We set a qualification test which contains several questions about a passage and the conversation. The testing questions cover all the three tasks (the ambiguity detection, providing a clarification question and replies). The qualification test is designed for making sure that selected workers understand our annotation guideline well. Mturk workers who get a full score can have access to our tasks. We will make the qualification test publically available upon publication.
>
> [Question on one clarification reply] Those cases mainly come from one clarification reply is highly possible than others or other possible replies are not provided in the passage. When annotators find the question is ambiguous and write those clarification questions, they aim to confirm their hypothesis (very likely to be true) rather than seeking an answer.
>
> [Question on decoding strategy] We followed the same decoding strategy as BART. We set beam size as 5 and remove duplicated trigrams in beam search.
>
> [Pointer to ShARC] Thank you for pointing out the missing reference. We included and discussed it in the revised manuscript. Although related, their work is different from ours since their reading passages are limited to rules and their clarification question aims to seek missing information rather than clarifying ambiguity.
>
> [1] Landis J R, Koch G G. The measurement of observer agreement for categorical data[J]. biometrics, 1977: 159-174.

---

### Official Review · Reviewer_gcnP · 2021-07-21
**A dataset for detection and clarification of ambiguity in conversational QA**

**Rating:** 5
**Confidence:** 3

**Review:**

In this paper a new dataset called Abg-CoQA is presented.
The aim of this dataset is to present a resource for dealing with ambiguity in conversational question answering.
The motivation for such a resource is well-written and convincing.

Overall, while I find the paper and dataset very interesting, I believe that some refinements (and especially more data) may boost the potential of this resource.

Pros:
* Most of the paper is very clear and easy-to-read
* The paper deals with  a very interesting and important problem, which is often ignored. This may drive future research on this topic.
* Task formulation (detection -> clarification question generation -> clarification-based answering) is solid and makes a lot of sense.
* The four ambiguity types presented by the authors form a nice analysis.

Cons:
* I'm concerned with the amount of data (800 questions for train, 100 for dev and 100 for test), for several reasons.
First, this seems like quite a few questions to learn from, especially for a hard task like ambiguity detection (which involves numerous kinds of reasoning, according to the authors). Second, due to the small size development and test data, determining the significance of improvements may prove difficult. Third, it affects the authors' analysis of the annotation's quality (i.e., I find it difficult to tell whether 68 questions are enough for determining the inter-annotator agreement)
* The section describing the models misses some details and can be improved. Specifically, I found it hard to understand the detection model's input-output formulation (what are the four answer types?). Also, why is the "ambiguous" token appended - How does it relate to the classification head?
Additionally, I found it a bit hard to understand the model for clarification-based question  answering. While this is a generation task, the authors use BERT and XLNet models, which are not suited for generation. A clarification of the exact setup is needed.

Comments:
* The paper discusses two types of ambiguity in the intro, but they seem very general and thus a bit hard to grasp. Also, the reader is referred to the wrong table (I believe that the authors meant to refer to table 2). An example described in-text may improve the clarity there.
* Later, a finer-grained analysis of four ambiguity types is introduced. However, the connection between these two "ambiguity type systems" isn't discussed.
* I believe that $CQ$ should be defined as $CQ_i$ (similar to $Q_i$), to support the case where several ambiguity questions were asked along the conversation.
* In Table 1, it's not clear what "ambiguous passage" is (i.e. column named #P under "Ambiguous").

---

> ### Author Response · Authors · 2021-07-31
> **Response to R3**
>
> [Concern with the amount of the data]
> - For the ambiguity detection task, models can learn from all of the 8.4k questions which is a reasonable size by comparing with other published datasets. For the other two tasks, clarification question generation and clarification-based question answering, a common way handling small training size is to pre-train the models on large scale datasets, e.g., CoQA, then fine-tune them on our dataset (as what we did). In addition, our dataset encourages the community to develop data-efficient models or to adapt some transfer learning methods.
>
> - In order to determine the significance of improvement, researchers can report p-value or z-value by some statistical tests. Those significance tests take the sample size into consideration, thus it will not be an issue with our dataset. Even using a large scale dataset, researchers generally report the significant test as well when the main contribution of the work is proposing a better model.
>
> - 68 samples are a reasonable size for computing the inter-rater agreement in our case [1][2].
>
> [Details in the section of Models] The formulation and input-output details of each task is described in Section 3 and Section 6 (revised). We addressed your question about the anwer types in footnote 1 of the revised manuscript. Considering answer types is commonly used in extraction-based models for question answering tasks. We append the “ambiguous” token at the end of the input passage so that the model is able to extract it as the prediction result (with the answer type of extraction). I hope footnote 1 also answers your question about the clarification-based question answering models: we consider the question answering task as a span extraction problem rather than a generation problem.
>
> [Ambiguity types] We addressed your comments in Section 5 of the revised manuscript.
>
> [#P in Table 1] Column named #P under "Ambiguous" shows the number of passages where at least one ambiguous question is identified in the conversation.
>
> [1] Sim J, Wright C C. The kappa statistic in reliability studies: use, interpretation, and sample size requirements[J]. Physical therapy, 2005, 85(3): 257-268.
>
> [2] Donner A, Eliasziw M. A goodness‐of‐fit approach to inference procedures for the kappa statistic: Confidence interval construction, significance‐testing and sample size estimation[J]. Statistics in medicine, 1992, 11(11): 1511-1519.

---

### Author Response · Authors · 2021-07-31
**General Response to all reviewers**

We thank the reviewers for their positive feedback (R1: “ a solid dataset paper”, “targets a very important research problem”; R2: “well-motivated and presented”, “follows most of the standard procedures expected”; R3: “well-written and convincing”, “ very interesting and important problem”) and insightful comments that helped us to significantly revise and improve our work.

We summarize the major points which we would like to clarify. The main contribution of our work is to propose a novel dataset for clarifying the ambiguity in conversational question answering. The ambiguity problem is challenging and we didn’t intend to solve it in this single work. Our work provides a first and solid step towards the problem by releasing a reliable dataset, conducting insightful analysis and reporting the performance of several state-of-the-art models. Our proposed challenge can easily be followed by further work, such as scaling up the dataset, proposing effective methods and evaluating reasoning models, etc.

Most of the reviewers pointed out their concern with the size of our dataset. However, the amount of data doesn’t weaken our contribution with the following reasons. First, our dataset is not really small. We annotated 8.6k conversational questions and 1k of them are ambiguous. For the task of ambiguity detection, models can be trained with all of the 8.6k questions. Second, our collected samples are a valuable contribution to the community. The annotations are quite expensive because the pipeline includes several tasks (ambiguity detection, providing clarification turns, answering questions after the clarification). The total annotation cost is more than 3k dollars. Third, even though our dataset is not on a large scale, it could be easily extended by further collection. We will make our annotation interface, coding manuals, etc., publically available.

With respect to the revised manuscript, we added a second baseline model for the ambiguity detection task (as requested by R2): the experimental result in Table 3, the related model description in Section 6 and the related discussion in Section 7.3.

---

### Decision · Program_Chairs · 2021-08-18

**Decision:**

Accept

**Comment:**

This paper presents a dataset for clarifying ambiguity in conversational questions based on the CoQA dataset. By hiding conversation turns and filtering based on QA performance, a set of more ambiguous questions can be pre-identified. The analysis discusses different clarification strategies and clarification subtasks, presenting models and evaluation for each. The reviewers agreed that this was an interesting problem and a well-written paper. The primary concerns are about shortcomings in the dataset, namely (a) the low amount of data (1k ambiguous questions out of 8.6k questions); (b) low agreement rate about ambiguity. These are serious concerns. However, the authors respond to this well. This is already a significant contribution in this area, and as a test set, this can help prove out (a) the viability of the annotation protocol; (b) establish a dataset for future work to build off of, even if it is imperfect. Moreover, settings like ambiguous questions are naturally going to have higher disagreement than more straightforward tasks like information-seeking QA, so I believe efforts in this area need to be judged accordingly.